# Neighborhood Deprivation Associated with Impaired Sit-to-Stand Performance in Middle-Aged and Older Adults: A Cross-Sectional Analysis with Clinical Implications

**DOI:** 10.3390/healthcare14010111

**Published:** 2026-01-02

**Authors:** Kenneth Harrison, Silvia Campos-Vargas, Brandon M. Peoples, Keven G. Santamaria-Guzman, David T. Redden, Michael A. Samaan, Jaimie A. Roper

**Affiliations:** 1School of Kinesiology, Auburn University, Auburn, AL 36849, USA; kdh0077@auburn.edu (K.H.); szc0259@auburn.edu (S.C.-V.); bmp0049@auburn.edu (B.M.P.); kgs0071@auburn.edu (K.G.S.-G.); dredden@auburn.vcom.edu (D.T.R.); 2Department of Kinesiology & Health Promotion, University of Kentucky, Lexington, KY 40506, USA; michael.samaan@uky.edu

**Keywords:** healthy aging, socioeconomic disparities, health equity, community-based research

## Abstract

**Highlights:**

This manuscript examines the relationship between area-level socioeconomic disadvantage and mobility health in community-dwelling adults (60 ± 16 years). Using the Area Deprivation Index (ADI) as a measure of neighborhood disadvantage, we investigate its association with performance on two key mobility assessments: the Instrumented Timed Up and Go (iTUG) test and the Instrumented Five Times Sit-to-Stand (i5TSTS) test.**What are the main findings?**Our findings reveal significant links between higher area deprivation and poorer mobility outcomes, particularly in lower limb strength and sit-to-stand transitions.**What are the implications of the main findings?**This study contributes to our understanding of how socioeconomic factors influence functional health in aging populations and has implications for developing targeted interventions to promote healthy aging in disadvantaged communities.

**Abstract:**

**Background**: Socioeconomic factors significantly influence health outcomes in older adults, yet their impact on specific aspects of mobility remains unclear. This study investigates the relationship between area-level socioeconomic disadvantage and mobility health in older adults. **Methods**: We conducted a cross-sectional study of 110 community-dwelling older adults recruited and collected using voluntary response sampling at eight health fairs across rural Southeast Alabama in 2022–2024 (60 ± 16 years, 80% female). Area-level socioeconomic status was measured using the Area Deprivation Index (ADI), a validated composite measure of neighborhood disadvantage. Mobility was assessed using the Instrumented Timed Up and Go (iTUG) test and the Instrumented Five Times Sit-to-Stand (i5TSTS) test. Kruskal–Wallis tests and general linear models in SAS 9.4 analyzed the relationship between ADI and mobility measures. **Results**: Higher ADI scores were significantly associated with poorer performance on the i5TSTS test (*p* = 0.0004). While overall iTUG duration did not differ significantly across ADI groups, the sit-to-stand phase of the iTUG showed a significant relationship with ADI (*p* = 0.0026). These associations remained significant after adjusting for age, weight, race, and education level. **Conclusions**: These findings suggest that neighborhood context plays a crucial role in mobility health, particularly in functions related to postural transitions. Clinicians should consider area-level disadvantage when screening for mobility limitations and may need to prioritize sit-to-stand interventions for patients living in high-deprivation areas.

## 1. Introduction

Mobility is a fundamental aspect of human life, shaping how individuals interact with their environment, perform activities of daily living, and participate in social engagements. Mobility health, which encompasses an individual’s ability to move safely and independently, is a key indicator of overall well-being in older adults [1,2,3]. Conversely, as mobility declines with age, an individual can experience variability in terrain and environmental conditions, leading to diminished engagement with their surroundings. These age-related changes have been observed to lead to reduced quality of life and increased risk of adverse health outcomes [4,5]. Consequently, the nuanced relationship with an individual’s environment is riddled with ambiguity due to various factors such as individual characteristics, neighborhood design, and access to services, shaping lifestyle behaviors.

Among the myriad environmental factors affecting mobility, socioeconomic conditions at the neighborhood level have emerged as a critical area of study. The Area Deprivation Index (ADI) is a composite measure to quantify and analyze these conditions. The ADI is an index that captures multiple dimensions of socioeconomic disadvantage, including income, education, employment, and housing conditions within a neighborhood [6,7]. Previous studies have shown that living in disadvantaged areas is associated with poorer health outcomes, including increased risk of chronic diseases, disability, and mortality [8,9,10,11,12]. Moreover, socioeconomic health disparities have widened in recent decades, with individuals living in disadvantaged neighborhoods experiencing worse health outcomes than those in more affluent areas [5,10,13,14,15,16]. However, the specific impact of neighborhood disadvantage on mobility health in older adults still needs to be studied. Investigating the effect of ADI on mobility health can illuminate the mechanisms underlying these disparities and guide efforts to reduce them.

Two widely used clinical assessments of mobility health in older adults are the instrumented Timed Up and Go (iTUG) test and the instrumented Five Times Sit-to-Stand (i5TSTS) test. The iTUG test measures postural transitions that are important in activities of daily living and navigating the environment [17]. Longer iTUG times have been associated with an increased risk of falls, disability, and mortality in older adults [18,19,20]. The i5TSTS test, on the other hand, assesses lower limb strength, which is essential for transitioning from sitting to standing and vice versa [21,22]. Slower i5TSTS times have been linked to decreased mobility, increased fall risk, and reduced independence in older adults [21,23,24]. However, little work has been done to understand the link between the level of socioeconomic disadvantage and performance in either of these tests. This gap in knowledge limits our understanding of how socioeconomic factors may differentially impact various aspects of mobility health, potentially hindering the development of targeted interventions to address mobility-related health disparities in older adults. Therefore, examining the relationship between ADI, iTUG, and i5TSTS performance can allow us to better understand how neighborhood socioeconomic status influences different aspects of mobility health.

Our aim is to investigate the relationship between socioeconomic disadvantages and mobility health outcomes in a sample of middle-aged to older adults in the Deep South. We hypothesize that higher levels of disadvantage will be associated with worse performance on the iTUG and i5TSTS [25]. Insights gained from this study can play a pivotal role in shaping public health policies, community-based interventions, and clinical practice guidelines that consider the influence of neighborhood context on promoting healthy aging and reducing health disparities.

## 2. Materials and Methods

### 2.1. Study Design

This research employed a cross-sectional observational design with retrospective analysis. Data were collected at community health fairs in the Deep South between Fall 2022 and Spring 2024, providing a snapshot of mobility health in relation to area deprivation. Participants were recruited using voluntary response sampling, and collections were performed on-site. The Instrumented Timed Up and Go (iTUG) test and the Instrumented Five Times Sit-to-Stand (i5TSTS) test were administered to assess mobility health, while demographic information was collected via questionnaire. The study protocol was approved by the Institutional Review Board of Auburn University (22-434 EX 2209). Since the study is based on minimal risk, a waiver for informed consent was granted. All participants, before participating, were provided with an IRB-approved information letter that detailed the study’s purpose, potential risks, and benefits.

### 2.2. Health Fairs

The goal was to supplement traditional in-lab collection and recruitment methods. Our team reached out to nearby communities to build relationships and secure tables or slots at health fairs. These fairs included themes like “prescription drug take-back day,” “Prostate Awareness Day,” and “Employee Wellness Day.” They focused on health literacy and offered connections and opportunities for those in rural areas to access healthcare they might not otherwise get. Most of these fairs targeted older adults. Our recruitment involved setting up a table/booth with brochures, flyers, and gear from our school to attract visitors. Anyone interested who stopped by was guided to the inclusion criteria information and a questionnaire. This approach provided a cost-effective and efficient way to explore the link between area-level socioeconomic factors and objective measures of mobility. Most importantly, it enabled face-to-face interaction between urban scientists and the rural populations we aim to serve, who might not be able to reach our campus or laboratory.

### 2.3. Participants

The study sample consisted of community-dwelling adults (60 ± 16 years) who attended health fairs in a variety of locations in southeastern Alabama during the years 2023–2024. Inclusion criteria were the ability to walk independently with or without an assistive device and residing within the study area. Exclusion criteria included inability to follow simple instructions or any acute medical condition that would preclude safe participation in the mobility tests. The ability to follow simple instructions was assessed informally through participants’ capacity to understand and provide informed consent and respond appropriately to the demographic questionnaire. Demographic characteristics, including mass, height, age, sex, education level, and self-reported health status, were collected via questionnaire to describe the sample and control for potential confounding factors in the analysis.

### 2.4. Collection Protocol

All participants followed standard testing procedures given by the same testers. Trials used for this analysis included the iTUG and i5TSTS, which were always performed as the first two trials in the same order. The same standard armless chair with a seat height of 43 cm was used for all testing. Due to the variety of health fairs, testing environments varied in terms of location, floor type (outdoor vs. gym floors), sound, and time of day. Data used in this analysis comes from a total of 8 different health fair collection sites. After reading the IRB-approved information letter detailing the study, participants first completed a demographic questionnaire. The participants then performed the iTUG, followed by the i5TSTS. Mobility tests were administered using wearable inertial sensors (APDM Mobility Lab V2, Portland, OR, USA) attached to the participant. Overall durations as well as iTUG phase durations were calculated and recorded automatically as seconds.

### 2.5. Inertial Measurement Unit (IMU)

Each participant was equipped with a standardized 6-sensor wearable inertial measurement system (APDM Mobility Lab V2, Portland, OR, USA). The sensors were placed at specific anatomical locations: (a) the sternum, positioned at the center of the flat surface of the chest below the point where the collarbone meets, (b) the lower back, centered on the lumbar region at the base of the spine, (c) on each wrist, worn like a watch, and (d) on each foot, centered on the top of the foot (Figure 1 and Figure 2). With this setup, the APDM system wirelessly synchronized all sensors and sampled data at 128Hz. Data was processed with the Mobility Lab V2 software (Moveo Explorer, APDM Inc., Portland, OR, USA). The Mobility lab V2, utilizing the iTUG assessment, measured the overall time in seconds to complete the task, postural transitions (sit-to-stand and stand-to-sit durations in seconds), turn angles (degrees of rotation), turn duration (time to complete the turn in seconds), turn velocity in m/s, and trunk lean angles (sit-to-stand and stand-to-sit in degrees). For the i5TSTS, the system recorded the total task time in seconds and the lean angles in degrees. All mobility assessments were performed by two trained PhD students who followed the same standardized script for each test. This sensor-based approach offers significant advantages over traditional observational methods by eliminating subjective bias and detecting subtle mobility impairments in movement quality, symmetry, and efficiency that are not visible through visual observation or stopwatch timing. The multi-sensor setup captures detailed accelerometer, gyroscope, and magnetometer data across multiple body segments, providing a comprehensive assessment of functional mobility beyond just temporal measures.

### 2.6. Measures

#### 2.6.1. Area Deprivation Index (ADI)

The Area Deprivation Index (ADI) is a validated composite measure of neighborhood-level socioeconomic disadvantage, incorporating indicators of income, education, employment, and housing quality. The ADI for each participant’s residential area was determined at the Census Block Group level using the Neighborhood Atlas tool developed by Kind et al. in 2018. The county location of each health fair was geocoded and matched to its corresponding Census Block Group using ADI data from the closest measured year. The national percentile rank of ADI was used, with higher percentiles indicating greater area deprivation. There were a total of 4 different counties where the health fairs occurred that we could attend.

#### 2.6.2. Instrumented Timed Up and Go (iTUG) Test

Participants were instructed to stand up from a standard armchair, walk 3 m at their normal pace, turn around, walk back to the chair, and sit down. The iTUG provided detailed metrics on sit-to-stand and stand-to-sit transitions, gait, and turning performance [17,18].

#### 2.6.3. Instrumented Five Times Sit-to-Stand (i5TSTS) Test

Participants were instructed to stand up fully and sit down five times as quickly as possible, with arms folded across the chest. The time from the initial seated position to the final seated position after five stands was recorded [21,23].

#### 2.6.4. Statistical Analysis

Statistical analyses were performed using SAS 9.4 software. Sample size was determined pragmatically based on the number of health fairs conducted and participant availability within the study timeframe. This community-engaged approach prioritized reaching underserved rural populations rather than predetermined sample targets. Post hoc sensitivity analyses confirmed adequate power (1 − β > 0.80) to detect the observed effect sizes for significant associations between ADI and mobility measures. Descriptive statistics were calculated for all variables. Analyses began by testing differences in mobility measures across ADI national groups using analysis of variance and multiple regression techniques. However, examination of the residuals using histograms and normal probability plots indicated a violation of the normality assumption. Due to the non-normal distribution of the residuals, non-parametric tests were employed. The Kruskal–Wallis test was used to compare the distributions of mobility measures (iTUG components and i5TSTS) across ADI national groups. A non-parametric regression (NP regression) approach, rank regression, was used to further examine the relationship between ADI and the rank of the mobility measures, with National ADI as the independent variable and the rank of various mobility parameters as dependent variables. The rank regression approach allowed for the analysis of ranked data and the inclusion of covariates. First, we used the Kruskal–Wallis test to compare distributions of mobility measures across ADI national groups. We then applied a rank-based general linear model approach (which functions as a non-parametric regression) to further examine these relationships while controlling for covariates. In this approach, we transformed both the dependent variables (mobility measures) and continuous independent variables (age, weight) into ranks before applying the GLM procedure. This rank-based GLM allowed us to analyze the data non-parametrically while still accommodating the inclusion of multiple covariates and categorical factors. Model results are reported with F-statistics, *p*-values, and R^2^ values derived from these rank-transformed data. Models were adjusted for potential confounding factors such as rank of age, rank of weight, race, and education level.

For each mobility measure, separate rank regression analyses were conducted to assess the impact of ADI while controlling for different sets of covariates. The significance of the overall model and individual predictors was evaluated, and parameter estimates were calculated to quantify the relationships. Statistical significance was set at *p* < 0.05 for all analyses. All tests were two-tailed.

## 3. Results

### 3.1. Participant Characteristics

The study involved 110 participants with ages ranging from 20 to 87, with an average age of 60 ± 16 years, including 89 females (80.9%, 173 ± 40 lbs.) and 21 males (19.1%, 191 ± 30 lbs). This sex imbalance reflects well-documented gender disparities in preventive health screening participation, with males demonstrating lower engagement in voluntary health assessments even when present at community health events. The race composition was comparable across both sex groups, with the majority of participants identifying as Black or African American (51.8%) or White or Caucasian (40%). Participants were divided into four groups based on their ADI index scores, which are ranked nationally for each county, with a higher ADI indicating greater levels of disadvantage: on-campus in Auburn, Alabama (ADI: 54, n = 13), Phenix City, Alabama (ADI: 60, n = 11), Opelika, Alabama (ADI: 81, n = 73), and Tuskegee, Alabama (ADI: 98, n = 13). The variation in sample sizes across locations reflects the differing number of health fairs in each area. Additionally, some events attracted larger community audiences, limiting the demographic data collected from those who completed gait assessments. Although this will be addressed in the Limitations section, it is also important to note this as a key finding.

### 3.2. i5TSTS, iTUG, and ADI

For overall analysis, the Kruskal–Wallis analysis indicated significant differences in i5TSTS performance across ADI groups (*p <* 0.01). In contrast to iTUG duration, the Kruskal–Wallis test did not show significant differences across ADI groups (*p* = 0.0865). However, when examining specific phases of the iTUG, such as the Sit-to-Stand phase, the Kruskal–Wallis test indicated significant differences across ADI groups (*p* = 0.0026) (Table 1).

The Kruskal-Wallis test revealed significant differences in i5TSTS performance across ADI groups (Chi-Square = 17.96, df = 3, *p* < 0.01). Participants from areas with higher ADI scores (more disadvantaged) generally performed worse on the i5TSTS test (Figure 3).

General linear model (GLM) analysis of i5TSTS, with ADI as the independent variable, showed a significant overall model (F = 7.42, *p* < 0.01, R^2^ = 0.21). The parameter estimates indicated that participants from areas with an ADI of 54 performed significantly better than those from areas with an ADI of 98 (estimate = −41.54, *p* < 0.01). When adjusting for age in the GLM, the model remained significant (F = 5.51, *p* < 0.01, R^2^ = 0.21), with ADI still showing a significant effect (*p* < 0.01). Notably, age itself was not a significant predictor (*p* = 0.8202) in the i5TSTS model, which may reflect our relatively restricted age range (60 ± 16 years), the health-motivated nature of health fair attendees who may represent a more physically active subset of their age cohort, or the possibility that neighborhood-level factors exert stronger influence on sit-to-stand performance than chronological age alone within this sample. Further GLM analyses adjusting for weight, race, and education level showed similar patterns, with ADI remaining a significant predictor of i5TSTS performance (Table 2).

For the overall iTUG duration, the Kruskal–Wallis test did not show significant differences across ADI groups (Chi-Square = 6.58, df = 3, *p* = 0.0865). Subsequent GLM analysis of overall iTUG duration, with ADI as the independent variable, showed a non-significant model (F = 2.23, *p* = 0.0905, R^2^ = 0.07). However, when examining specific phases of the iTUG such as the Sit-to-Stand phase, the Kruskal–Wallis test showed significant differences across ADI groups (Chi-Square = 14.2604, df = 3, *p* = 0.0026). The GLM analysis confirmed this finding (F = 5.30, *p* = 0.0022, R^2^ = 0.16). The stand-to-sit phase showed no significant differences across ADI groups (Kruskal–Wallis Chi-Square = 4.6548, df = 3, *p* = 0.1989), as well as the walking phase. The turning phase had marginally significant differences found across ADI groups (Kruskal–Wallis Chi-Square = 7.9985, df = 3, *p* = 0.0460). When using GLM analyses to adjust for covariates such as age, weight, race, and education, the results generally remained consistent, with the sit-to-stand phase showing the strongest and most consistent relationship with ADI.

## 4. Discussion

The objective of this study was to investigate the relationship between area-level socioeconomic disadvantages (Area Deprivation Index, ADI) and mobility health (i5TSTS and iTUG) in community-dwelling adults. Our findings reveal two key results. First, there was a significant association between higher ADI scores and poorer performance on the i5TSTS test, indicating that older adults from more disadvantaged areas demonstrated reduced lower limb strength and function. Second, while the overall iTUG duration did not significantly differ across ADI groups, the sit-to-stand phase of the iTUG showed a similar pattern to the i5TSTS results, with participants from higher ADI areas performing worse. These findings suggest that area-level socioeconomic factors may have a specific impact on lower limb strength and the ability to rise from a seated position. However, our sample ADI range (54–98 national percentile) reflects the socioeconomic reality of the Deep South, where few neighborhoods have levels of disadvantage that fall below the national median. These findings describe gradients within a disadvantaged region and may not generalize to more affluent areas of the country.

This selective vulnerability suggests that sit-to-stand transitions, which are heavily dependent on lower limb strength, particularly quadriceps and gluteal muscle power, are especially susceptible to the environmental and resource factors that characterize disadvantaged neighborhoods. In contrast, overall iTUG duration incorporates gait speed and turning, which may be more resilient to neighborhood-level influences or rely on different physiological systems. The mechanisms we discuss throughout this section, including limited physical activity infrastructure, nutritional inadequacies, chronic stress, and healthcare access barriers, appear to converge most directly on muscle-dependent postural transitions rather than affecting all mobility domains equally. This finding has important clinical implications, as the sit-to-stand function serves as both an early marker of functional decline and a specific intervention target for older adults in disadvantaged areas.

Our findings align with and extend prior research on socioeconomic disparities in functional health. The significant association between ADI and sit-to-stand performance is consistent with Thorpe et al.’s work demonstrating that socioeconomic resources predict late-life mobility decline, and Coppin et al.’s findings linking low socioeconomic status to physiological impairments in older adults [9,10]. Our observed effect sizes (R^2^ = 0.16–0.21 for significant associations) are comparable to those reported in similar community-based studies examining environmental determinants of mobility. However, our specific finding that sit-to-stand transitions are more vulnerable to neighborhood disadvantage than overall mobility duration adds nuance to the literature, suggesting that socioeconomic factors may differentially impact strength-dependent versus endurance-dependent mobility components.

The selective association between ADI and sit-to-stand performance in disadvantaged rural communities has important implications. Postural transitions are fundamental to maintaining independence in aging populations [1,2,26,27] and represent critical functional movements essential for daily living. The i5TSTS and sit-to-stand phase of the iTUG, which our study found to be significantly associated with ADI, are not merely isolated physical tasks but represent critical functional movements essential for daily living. These postural transitions require a complex integration of lower limb strength, balance, and motor control [22,24,27,28]. Moreover, the i5TSTS and iTUG tests have been widely validated as clinical tools for assessing fall risk, functional decline, and overall mobility health in older adults [18,22,29,30].

Several mechanisms may explain the observed relationship between ADI and reduced lower limb function. Disadvantaged areas often lack physical activity resources (parks, walking paths, fitness facilities) [31], limiting engagement in weight-bearing exercises crucial for lower limb strength [9]. Nutritional inadequacies are common in areas with limited access to nutrient-dense foods, impacting muscle health [32]. Chronic stress in disadvantaged neighborhoods elevates cortisol levels, accelerating muscle catabolism [33]. Finally, reduced healthcare access may result in poorer management of mobility-affecting conditions such as arthritis or diabetes [34,35]. These interrelated factors likely create a complex web of influence that collectively could correspond to the observed disparities in lower limb function and sit-to-stand performance across ADI levels.

Limited physical activity infrastructure in high-ADI areas presents a significant barrier to mobility health. The absence of safe walking surfaces, adequate lighting, and age-appropriate exercise equipment deters older adults from regular physical activity [36]. This infrastructural deficit can significantly impact the ability of older adults to engage in regular exercise, particularly activities that promote lower limb strength and function. For instance, the absence of level walking surfaces, adequate lighting, or benches for rest can deter older adults from engaging in walking or outdoor exercises [37,38]. Limited transportation access further restricts the ability to reach available resources [39,40], creating a cycle of inactivity that contributes to observed disparities in sit-to-stand performance.

Nutritional factors are associated with maintaining mobility health, and areas with higher ADI scores often face significant challenges in accessing adequate nutrition. Disadvantaged neighborhoods frequently exhibit characteristics of “food deserts,” with limited access to affordable, nutrient-dense foods and an overabundance of fast-food outlets and convenience stores [41]. This nutritional environment could lead to deficiencies that directly impact muscle health and function in older adults. Particularly concerning is the often-inadequate protein intake observed in low-income areas, which can accelerate sarcopenia—the age-related loss of muscle mass and function [42]. Sufficient protein intake is essential for muscle protein synthesis and the maintenance of muscle mass, strength, and function in older adults [43]. Moreover, micronutrient deficiencies, such as vitamin D and calcium, which are common in areas with limited access to fresh produce and dairy products, can negatively affect bone health and muscle function [44]. The compounding effect of these nutritional inadequacies may manifest in reduced muscle strength and impaired functional performance, particularly in weight-bearing activities like the sit-to-stand motion. Additionally, higher prevalence of obesity in disadvantaged areas, often a result of poor nutritional environments, can further compromise mobility by increasing mechanical load on joints and exacerbating sarcopenia through increased inflammation [45]. The intricate relationship between neighborhood deprivation, nutritional access, and physiological factors underscores the potential for nutrition to be a significant mediator in the observed association between higher ADI and reduced lower limb function in older adults.

Chronic health conditions, which are more prevalent in the Southern United States and often associated with areas of higher deprivation, present another crucial mechanism linking ADI to reduced mobility in older adults. The “Stroke Belt,” encompassing much of the southeastern U.S., exemplifies this regional health disparity, with higher rates of stroke, hypertension, and diabetes compared to national averages [46]. These conditions, along with others such as obesity and arthritis, are more common in areas with higher ADI and can significantly impact mobility [25]. Diabetes, for instance, can lead to peripheral neuropathy and reduced lower extremity function, directly affecting an individual’s ability to perform sit-to-stand movements [47]. Similarly, the higher prevalence of osteoarthritis in the South, exacerbated by factors such as obesity and physical inactivity common in disadvantaged areas, can cause joint pain and stiffness, further limiting mobility [8]. Cardiovascular diseases, including hypertension and heart failure, can reduce exercise capacity and contribute to muscle weakness, impacting overall mobility [40]. Moreover, the management of these chronic conditions often requires regular medical care, which may be less accessible or of lower quality in areas with high ADI, particularly in rural Southern regions [5,34,35]. This combination of higher disease prevalence and potential undermanagement in disadvantaged areas of the South creates a perfect storm for mobility impairment, potentially explaining the strong association observed between ADI and reduced lower limb function in our study.

### Limitations

Our study provides valuable insights into the relationship between area-level socioeconomic disadvantages and mobility health in middle-aged and older adults, but several limitations should be acknowledged when interpreting our findings. The underlying goal of this research project was to lean into some of the limitations as a strategy to cause a net improvement in scientific rigor and ecological validity. Sacrificing some of the control that comes with in-lab collections, for the benefit of improving sample demographics and access to rural populations, is a strategy that can potentially complement current practices, not replace them.

Inherent design limitations: The cross-sectional design of our study prevents us from establishing causal relationships between area deprivation and mobility outcomes. We cannot determine whether the observed associations reflect the impact of current neighborhood conditions, cumulative effects of long-term exposure to disadvantaged environments, or other factors that may influence both residential location and mobility health. Longitudinal studies tracking changes in both neighborhood characteristics and mobility measures over time would provide stronger evidence for causal relationships. Additionally, we did not collect data on the duration of residency in participants’ current neighborhoods. The relationship between ADI and mobility health may vary based on how long individuals have been exposed to their current neighborhood environments. Long-term residents may experience different cumulative effects compared to those who recently moved into an area. Future studies should assess residential history to better understand the temporal dynamics of this relationship.

Methodological choices and their implications: The relatively small sample size (n = 110) limits the statistical power of our analyses, particularly when subdividing participants into four ADI groups, which resulted in unbalanced group sizes (ranging from n = 11 to n = 73). This imbalance reflects our pragmatic, community-engaged approach that prioritized access to underserved populations over predetermined sampling targets, and may affect the robustness of our between-group comparisons and potentially limit our ability to detect more subtle associations between ADI and mobility measures. Our recruitment through health fairs introduces a potential selection bias. Individuals who attend health fairs may be more health-conscious or have greater health literacy than the general population. This self-selection could result in better baseline mobility performance across all ADI groups, potentially underestimating the true disparities in the broader population. Conversely, health fairs in disadvantaged areas may attract individuals with existing health concerns, potentially amplifying observed associations. Without comparative demographic data from the surrounding populations, we cannot quantify the magnitude or direction of this selection bias. However, our community-engaged approach enabled us to reach rural populations who rarely participate in laboratory-based research, trading some internal validity for improved ecological validity and sample diversity.

Our use of county-level ADI based on health fair locations, rather than participants’ specific home addresses, represents a potential source of misclassification bias. This approach may have attenuated the true associations between neighborhood disadvantage and mobility outcomes, as within-county variation in socioeconomic conditions would not be captured. Additionally, health fair attendance could vary systematically across different socioeconomic areas, potentially over-representing certain demographic groups. While our approach successfully reached populations in rural areas who might not otherwise participate in research, the generalizability of our findings to the broader population of older adults, particularly those who are homebound or have severe mobility limitations, remains uncertain. This underscores the critical importance of meeting people where they are when trying to do research generalizable to an entire clinical population like older adults. Finally, mobility assessments were conducted by two trained PhD student raters following standardized protocols, though we did not calculate formal inter-rater reliability. The objective, sensor-based nature of our measurements minimizes subjective variability.

Unmeasured mechanisms: While our study documented significant associations between ADI and mobility measures, we did not directly assess the mechanisms underlying these relationships. Our discussion proposes several potential pathways, including limited access to physical activity resources, nutritional factors, and chronic health conditions, but we did not directly measure these mediating factors. Future research incorporating measures of physical activity levels, nutritional status, and detailed health histories would help elucidate the mechanisms linking neighborhood disadvantage to mobility health.

Despite these limitations, our study offers important insights into the relationship between area-level socioeconomic factors and mobility health in aging populations, particularly in the Deep South, a region often underrepresented in research. By acknowledging these limitations and suggesting directions for future research, we hope to contribute to a more comprehensive understanding of how neighborhood contexts shape healthy aging and mobility.

## 5. Conclusions

Our findings demonstrate that neighborhood-level socioeconomic disadvantage is significantly associated with impaired sit-to-stand performance in middle-aged and older adults living in disadvantaged rural communities, even after controlling for individual-level factors. This selective vulnerability of lower limb strength suggests that postural transitions represent both an early functional biomarker and a critical intervention target for addressing health disparities. These results support several actionable public health strategies: (1) implementing ADI-stratified mobility screening at community health fairs and primary care clinics to identify at-risk older adults in disadvantaged areas; (2) deploying evidence-based strength training programs (e.g., Otago Exercise Program, sit-to-stand protocols) through partnerships with community centers in high ADI neighborhoods; (3) advocating for municipal infrastructure investments, such as outdoor fitness equipment, benches, and safe walking paths, prioritized for disadvantaged areas; and (4) integrating neighborhood socioeconomic data into fall risk assessment protocols.

By recognizing neighborhood context as a determinant of mobility health in disadvantaged rural communities, clinicians and policymakers can develop more equitable, targeted interventions to promote healthy aging across diverse socioeconomic backgrounds.

## Figures and Tables

**Figure 1 healthcare-14-00111-f001:**
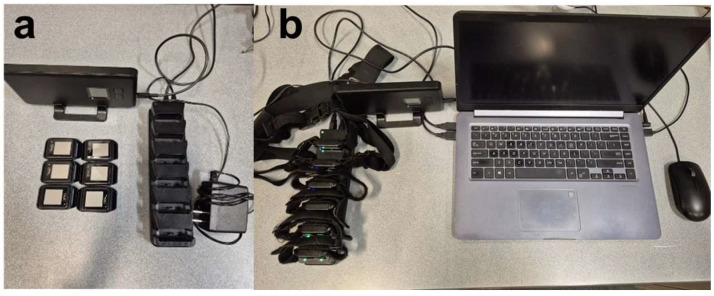
APDM mobility lab instrumentation system: (**a**) six tri-axial IMUs with charging docks and a wireless access point for synchronized data transmission; (**b**) data acquisition workstation with six tri-axial IMUs, bands, and the ready-to-run Moveo Explorer V2 software for real-time data extraction (original image).

**Figure 2 healthcare-14-00111-f002:**
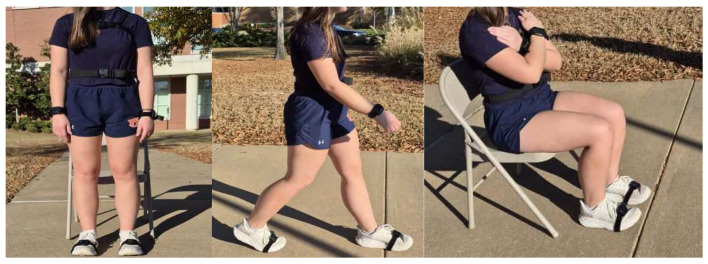
Six-sensor IMU configuration with standardized anatomical placement for iTUG and i5TSTS functional mobility assessments (original image).

**Figure 3 healthcare-14-00111-f003:**
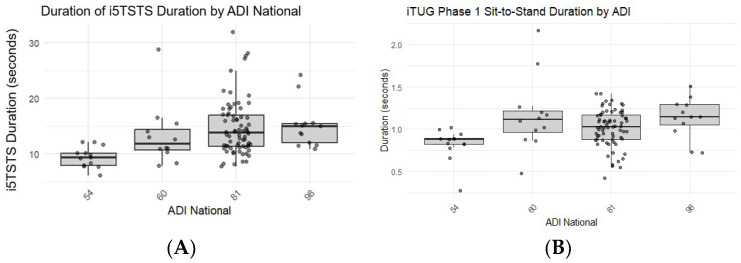
Boxplot showing the distribution of mobility test performance (durations in seconds) across different levels of disadvantage (ADI): (**A**) i5TSTS duration by ADI National. (**B**) iTUG duration of the sit-stand phase by ADI National. Higher ADI values indicate higher levels of disadvantage.

**Table 1 healthcare-14-00111-t001:** *p*-values of association between outcome and ADI National, adjusting for covariates. Sit to Stand Duration, Tug Duration, and, more specifically, the Sit to Stand phase of the TUG, were the only variables that remained significant when including race and education covariates.

Covariates	Sit to Stand Duration	Tug Duration	Tug Sit to Stand	Tug Walking	Tug Turn	Tug Stand to Sit
ADI National	0.0004	0.0865	0.0026	0.2746	0.0460	0.1989
ADI National + Age	0.0002	0.1349	0.0071	0.2324	0.4306	0.2171
ADI National + Weight	0.0003	0.0798	0.0019	0.3476	0.0599	0.2178
ADI National + Race	0.0086	0.0216	0.0006	0.0898	0.0470	0.1701
ADI National + Education	0.0095	0.1092	0.0034	0.2260	0.2175	0.4115
ADI National + Race + Education	0.0383 *	0.0180 *	0.0009 *	0.0545	0.1776	0.2815

* *p* < 0.05.

**Table 2 healthcare-14-00111-t002:** General linear model results for key measures.

Mobility Measure	Model Covariates	F-Statistic	*p*-Value	R^2^
i5TSTS Duration	ADI National	7.42	<0.01	0.21
i5TSTS Duration	ADI National + Age	5.51	<0.01	0.21
iTUG Duration	ADI National	2.23	0.0905	0.07
iTUG Sit-to-Stand Phase	ADI National	5.30	0.0022	0.16

## Data Availability

The data presented in this study are available upon request from the corresponding author.

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
