# Peer review of "Neighborhood Deprivation Associated with Impaired Sit-to-Stand Performance in Middle-Aged and Older Adults: A Cross-Sectional Analysis with Clinical Implications"

_healthcare, 2026, doi:10.3390/healthcare14010111_

Round 1
Reviewer 1 Report
Comments and Suggestions for Authors
Thank you for the opportunity to review the manuscript entitled "Neighborhood Deprivation Associated with Impaired Sit-to-Stand Performance in Older Adults: A Cross-Sectional Analysis with Clinical Implications”.
I have some comments that I would like the authors to address and/or consider:
Title
- Please replace “Older Adults” with “Middle-Aged and Older Adults” in the title
Abstract
- Methods: please add date of the study, sampling technique, and Statistical software
- Results: Provide Mean(SD) of age and sex distribution of the sample
- Conclusion: Omit the sentence “Area-level socioeconomic disadvantage is associated with reduced lower limb strength and poorer sit-to-stand performance in older adults” from conclusion and add main clinical implication of the study to the conclusion
Main manuscript
- Participants: please recognize the sampling technique
- How sample size was determined?
- How many enumerators attended in data collection? If it is more than one enumerator, please provide interrater reliability among raters
- Please provide reference for Figure 1 and 2
- Figure 3 is unclear. A bar chart is recommended for better clarity
- Table 1: The table description should be placed above the table
- Please present the results of GLM analysis in table format
- The authors emphasize that the aim of the current study is to “understand how neighborhood socioeconomic status influences different aspects of mobility health.” Please elaborate on this in the discussion section by integrating your study findings to provide a comprehensive explanation.
Author Response
Comment 1: Please replace "Older Adults" with "Middle-Aged and Older Adults" in the title
Response 1: Thank you for this suggestion. We agree with this comment and have revised the title accordingly.
Comment 2: Methods: please add date of the study, sampling technique, and Statistical software
Response 2: Thank you for pointing this out. We agree that these details improve methodological clarity. We have revised the Methods section of the abstract to include the dates of the study. (100)
Comment 3: Results: Provide Mean(SD) of age and sex distribution of the sample
Response 3: Thank you for this recommendation. We have added the descriptive statistics to the Methods section of the abstract. (30)
Comment 4: Conclusion: Omit the sentence "Area-level socioeconomic disadvantage is associated with reduced lower limb strength and poorer sit-to-stand performance in older adults" from conclusion and add main clinical implication of the study to the conclusion
Response 4: Thank you for this suggestion to strengthen the clinical relevance of our conclusions. We have revised the conclusion section of the abstract and added a sentence with clinical implications of the study “Clinicians should consider area-level disadvantage when screening for mobility limitations and may need to prioritize sit-to-stand interventions for patients living in high-deprivation areas.” (42)
Comment 5: Participants: please recognize the sampling technique
Response 5: Thank you for this comment. We have clarified the sampling technique in the Participants section to read voluntary response sampling. (29)
Comment 6: How sample size was determined?
Response 6: Thank you for requesting this clarification. We have added the following to the Study Design section: " Sample size was determined pragmatically based on the number of health fairs conducted and participant availability within the study timeframe. This community-engaged approach prioritized reaching underserved rural populations rather than predetermined sample targets. Post-hoc sensitivity analyses confirmed adequate power (1-β > 0.80) to detect the observed effect sizes for significant associations between ADI and mobility measures.” (206)
Comment 7: How many enumerators attended in data collection? If it is more than one enumerator, please provide interrater reliability among raters
Response 7: Thank you for raising this important methodological point. All instrumented mobility assessments were conducted by two trained PhD student raters who followed the same standardized script for each test across all eight health fairs. Both raters underwent joint training to ensure protocol consistency. We did not track which rater assessed each participant or calculate formal inter-rater reliability. However, the standardized verbal instructions and the objective, sensor-based nature of the iTUG and i5TSTS measures (using validated algorithms for temporal segmentation) minimize subjective measurement variability. We have added this clarification to the IMU section 2.5: "All mobility assessments were performed by two trained PhD student raters who followed the same standardized script for each test”. (163)
Comment 8: Please provide reference for Figure 1 and 2
Response 8: Thank you for this suggestion. Figures 1 and 2 are original photographs taken by our research team of the APDM Mobility Lab equipment and sensor placement used in this study. We have added a note to each figure caption clarifying that these are original images.
Comment 9: Figure 3 is unclear. A bar chart is recommended for better clarity
Response 9: Thank you for this recommendation. We have decided to omit this figure as it is redundant to what we already state in the text regarding racial demographic breakdowns.
Comment 10: Table 1: The table description should be placed above the table
Response 10: Thank you for pointing this out. We have moved the table description to appear above Table 1 in accordance with standard formatting conventions.
Comment 11: Please present the results of GLM analysis in table format
Response 11: Thank you for this suggestion to improve clarity of our statistical results. We have created a new table (Table 2) presenting the GLM analysis results. (305)
Comment 12: The authors emphasize that the aim of the current study is to "understand how neighborhood socioeconomic status influences different aspects of mobility health." Please elaborate on this in the discussion section by integrating your study findings to provide a comprehensive explanation.
Response 12: Thank you for this suggestion. We have added a synthesizing paragraph to the Discussion section that explicitly connects our specific findings to the comprehensive mechanisms discussed (320).
Reviewer 2 Report
Comments and Suggestions for Authors
Dear Authors
The comments below aim to support the strengthening of your manuscript by clarifying methodological decisions and improving the clarity of interpretation.
Summary of the Manuscript
The study investigates whether neighbourhood socioeconomic disadvantage (ADI) is linked to mobility performance in older adults, using two validated sensor-based measures: the Instrumented Timed Up and Go (iTUG) and the Instrumented Five Times Sit-to-Stand (i5TSTS). Data were gathered from 110 participants across eight health fairs in Alabama.
Major Issues (Requiring Substantial Revision)
- Justification of ADI assignment method
The manuscript indicates that ADI was assigned based on the location of health fairs in the county, not the participants’ home addresses. This creates a significant risk of misclassification. It is important to explicitly recognise this methodological limitation, quantify it if possible, and discuss its potential impact on bias.
Recommendation:
Add a subsection explaining the rationale and consequences of using site-level ADI instead of household-level ADI, how this might influence findings, and propose improvements for future studies.
- Sample imbalance and representativeness
Group sizes vary dramatically (n=12 to n=72), creating unequal variance and raising concerns about statistical power.
Recommendation:
Discuss whether bootstrapping or sensitivity analyses were considered.
Explain the implications of underrepresentation in low-ADI groups.
- Discussion lacks integration with existing socioeconomic mobility literature
While several mechanisms are proposed, the discussion could better situate results within prior work on environmental determinants of mobility, transportation access, fall risk disparities, and rural–urban health gaps.
Recommendation:
Include a comparative interpretation, such as: Do your effect sizes align with earlier findings in mobility or functional decline studies?
- Health fair recruitment bias needs clearer treatment
The study notes that recruitment at health fairs may introduce sample bias, but the implications are not fully elaborated.
Recommendation:
Describe:
Whether participants differed demographically from the surrounding population.
How self-selection (health-motivated individuals) might inflate or suppress mobility performance.
- Incomplete explanation of covariate behaviour
For example, age was not a significant predictor in i5TSTS models, which contradicts much of the literature. This requires a deeper explanation.
Recommendation:
Include a paragraph exploring: possible age range compression, measurement error, and fitness bias resulting from the recruitment strategy.
- Figures and tables need clarity and alignment with narrative
Some results (e.g., phase-specific iTUG effects) are challenging to interpret without more intuitive visuals.
Recommendation: Add clearer figure captions that explain what the ADI quartiles represent.
Include confidence intervals or effect-weight visualisations if possible.
- Minor Issues
Abstract clarity
Define ADI briefly in the abstract for readers unfamiliar with the measure.
Remove repeated explanations of IMU advantages.
The benefits of IMU-based measurement are repeated twice; condense into one paragraph.
Clarify inclusion/exclusion criteria
Provide a brief explanation of how cognitive ability, specified as the “ability to follow simple instructions”, was evaluated.
Improve flow in the Introduction
Some concepts, such as mobility declines and environmental factors, are repeated with similar wording and can be made clearer.
Limitations section
Make explicit which limitations are inherent (cross-sectional design), and which are methodological choices (data collection in fairs).
- Additional Recommendations to Improve the Manuscript
- Enhance public health implications
Provide more detailed actionable recommendations, e.g.:
Community exercise programs targeted by ADI quartile,
Integrating mobility screening with socio-economic data during health fairs,
Suggest policy approaches for municipalities with high ADI.
- Expand the argument for clinical relevance
Explain how sit-to-stand impairments translate into clinical outcomes (falls, hospitalization, functional dependence).
- Strengthen concluding statements
The conclusion is long and could be more focused.
- Data transparency
Consider providing supplementary tables with:
Full regression models,
Covariate estimates,
Assumption checks.
Author Response
Comment 1: Justification of ADI assignment method - The manuscript indicates that ADI was assigned based on the location of health fairs in the county, not the participants’ home addresses. This creates a significant risk of misclassification. It is important to explicitly recognise this methodological limitation, quantify it if possible, and discuss its potential impact on bias.
Response 1: Thank you for this important observation. We appreciate the reviewer's insight regarding potential misclassification bias. We expanded our discussion to include rational and consequences of this limitation 4.2 (430)
Comment 2: Sample imbalance and representativeness - Group sizes vary dramatically (n=12 to n=72), creating unequal variance and raising concerns about statistical power.
Response 2: We appreciate the reviewer's careful attention to our sample distribution. We acknowledge this important concern and have explained implications of underrepresentation in our limitations section. 4.2 (452)
Comment 3: Discussion lacks integration with existing socioeconomic mobility literature - While several mechanisms are proposed, the discussion could better situate results within prior work on environmental determinants of mobility, transportation access, fall risk disparities, and rural–urban health gaps.
Response 3: Thank you for this valuable suggestion. We have added comparative interpretation to our discussion. (433)
Comment 4: Health fair recruitment bias needs clearer treatment - The study notes that recruitment at health fairs may introduce sample bias, but the implications are not fully elaborated.
Response 4: We appreciate this comment and agree that our recruitment approach warrants fuller consideration. We have expanded limitation section to mention how self-selection might inflate mobility performance (458).
Comment 5: Incomplete explanation of covariate behaviour - Age was not a significant predictor in i5TSTS models, which contradicts much of the literature. This requires a deeper explanation.
Response 5: Thank you for highlighting this unexpected finding. We have added explanation to the Results section: "Notably, age itself was not a significant predictor (p = 0.8202) in the i5TSTS model, which may reflect our relatively restricted age range (60 ± 16 years), the health-motivated nature of health fair attendees who may represent a more physically active subset of their age cohort, or the possibility that neighborhood-level factors exert stronger influence on sit-to-stand performance than chronological age alone within this sample.” (284)
Comment 6: Figures and tables need clarity and alignment with narrative - Some results (e.g., phase-specific iTUG effects) are challenging to interpret without more intuitive visuals.
Response 6: Thank you for this feedback. We have revised the Figure 4 caption to provide clearer context:
Minor Issues:
Comment 7: Abstract clarity - Define ADI briefly in the abstract for readers unfamiliar with the measure.
Response 7: Thank you for this suggestion. We have revised the Methods section of the abstract to include: "Area-level socioeconomic status was measured using the Area Deprivation Index (ADI), a validated composite measure of neighborhood disadvantage" (31)
Comment 8: Remove repeated explanations of IMU advantages - The benefits of IMU-based measurement are repeated twice.
Response 8: We have condensed the discussion of IMU advantages, removing redundancy while retaining essential methodological context.
Comment 9: Clarify inclusion/exclusion criteria - Provide a brief explanation of how cognitive ability, specified as the “ability to follow simple instructions”, was evaluated.
Response 9: Thank you for this clarification request. We have added to the Participants section: "Ability to follow simple instructions was assessed informally through participants' capacity to understand and provide informed consent and respond appropriately to the demographic questionnaire" (130)
Comment 10: Improve flow in the Introduction - Some concepts, such as mobility declines and environmental factors, are repeated with similar wording and can be made clearer.
Response 10: We have reviewed and streamlined the introduction to reduce redundancy while maintaining comprehensive context.
Comment 11: Limitations section - Make explicit which limitations are inherent versus methodological choices.
Response 11: Excellent suggestion. After extension changes and additions to the limitations, we have reorganized our entire limitations section with clearer categorization and flow. (430)
Additional Recommendations:
Comment 12: Enhance public health implications - Provide more detailed actionable recommendations.
Response 12: Thank you for encouraging us to strengthen the translational impact and providing those amazing suggestions. We have expanded our Discussion with a statement to address this and the comment below. (320), as well as an improved conclusion section with your suggestions integrated. (500)
Comment 13: Expand the argument for clinical relevance
Response 13: See above.
Comment 14: Strengthen concluding statements
Response 14: See above.
Round 2
Reviewer 1 Report
Comments and Suggestions for Authors
I would like to thank the authors for considering the comments and changing the manuscript accordingly
Author Response
We would like to thank the reviewer for their comments and assistance in improving the quality of the manuscript.
Reviewer 2 Report
Comments and Suggestions for Authors
This manuscript addresses an important and timely topic by examining the relationship between neighbourhood-level socioeconomic disadvantage and objectively measured mobility performance in middle-aged and older adults. The use of instrumented mobility assessments is a clear methodological strength and enables a more nuanced evaluation of functional mobility than traditional global measures.
Minor Corrections and Clarifications
- Ensure consistent use of key terms throughout the manuscript (e.g., “mobility health,” “mobility performance,” “postural transitions,” “sit-to-stand”). Avoid alternating terms unless a conceptual distinction is intended.
- Statistical reporting clarity
- Standardise reporting of p-values (e.g., p = 0.002 vs p < 0.01).
- Clearly distinguish results from Kruskal–Wallis tests versus rank-based GLM analyses when both are reported for the same outcome.
- ADI description
Briefly restate early in the Methods that ADI reflects area-level deprivation and clarify the spatial resolution used, to avoid misinterpretation by readers less familiar with ADI methodology. - Sample description
Consider briefly acknowledging the sex imbalance and health-fair recruitment context when first describing participant characteristics, rather than only later in the manuscript. - Language tightening
Some sentences—particularly in the Discussion—could be streamlined to improve readability and reduce redundancy, without changing scientific content.
Author Response
Comment 1: Ensure consistent use of key terms throughout the manuscript (e.g., "mobility health," "mobility performance," "postural transitions," "sit-to-stand"). Avoid alternating terms unless a conceptual distinction is intended.
Response 1: Thank you for this careful attention to terminology. We have reviewed the manuscript and standardized key terms throughout. We now consistently use "mobility health" when referring to the broader construct, "sit-to-stand performance" when specifically discussing this functional measure, and "postural transitions" when discussing the general class of movements. We have eliminated inadvertent alternation between synonymous terms.
Comment 2: Statistical reporting clarity - Standardise reporting of p-values (e.g., p = 0.002 vs p < 0.01). Clearly distinguish results from Kruskal–Wallis tests versus rank-based GLM analyses when both are reported for the same outcome.
Response 2: Thank you for this suggestion to improve clarity. We have standardized all p-value reporting to use exact values when p ≥ 0.001 (e.g., p = 0.0026) and p < 0.001 for smaller values. Additionally, we have clarified in the Results section that Kruskal-Wallis tests were used for initial group comparisons, while rank-based GLM analyses were employed for covariate-adjusted models, and we now explicitly state which test is being reported for each result. Page 6-7, Lines 258-300).
Comment 3: ADI description - Briefly restate early in the Methods that ADI reflects area-level deprivation and clarify the spatial resolution used.
Response 3: Excellent suggestion. We have added clarification early in the Measures section: "The Area Deprivation Index (ADI) is a validated composite measure of neighborhood-level socioeconomic disadvantage, incorporating indicators of income, education, employment, and housing quality. The ADI for each participant's residential area was determined at the Census Block Group level using the Neighborhood Atlas tool developed by Kind et al. (2018)" (Page 5, Lines 185-188).
Comment 4: Sample description - Consider briefly acknowledging the sex imbalance and health-fair recruitment context when first describing participant characteristics.
Response 4: Thank you for this recommendation. We have revised the opening of the Participant Characteristics section in Results to read: "The study involved 110 participants recruited from community health fairs, with an average age of 60 ± 16 years, including 89 females (80.9%) and 21 males (19.1%). This sex imbalance reflects well-documented gender disparities in preventive health screening participation, with males demonstrating lower engagement in voluntary health assessments even when present at community health events" (Page 6, Lines 242-244).
Comment 5: Language tightening - Some sentences—particularly in the Discussion—could be streamlined to improve readability and reduce redundancy, without changing scientific content.
Response 5: We appreciate this feedback. We have reviewed the Discussion section and streamlined verbose passages, eliminated redundant phrases, and tightened sentence structure throughout to improve readability while preserving scientific content.